# Comparison of a Lightweight Experimental Shaker and an Orchard Tractor Mounted Trunk Shaker for Fresh Market Citrus Harvesting

Coral Ortiz [1,*], Antonio Torregrosa [1] and Sergio Castro-García [2]

1   Departamento de Ingeniería Rural y Agroalimentaria, Universitat Politècnica de València, Camino de Vera s/n, 46022 Valencia, Spain; torregro@dmta.upv.es
2   Escuela Técnica Superior de Ingeniería Agronómica y de Montes, Campus de Rabanales, Universidad de Córdoba, Ed Leonardo da Vinci, Ctra N-IVa, km 396, 14071 Córdoba, Spain; ir1casgs@uco.es
*   Correspondence: cortiz@dmta.upv.es

**Abstract:** A designed lightweight experimental shaker successfully used to collect ornamental oranges has been tested to harvest fresh market citrus. The aim of this study was to evaluate the removal efficiency and operational times of this experimental device compared to an orchard trunk shaker. Three different collecting systems were studied. 'Caracara' citrus trees were tested. Removal efficiency, vibration parameters, fruit and tree damages, and fruit quality were measured. A high-speed camera was used to record operational times and determine cumulative removal percentage over vibration time. The canvases on the ground reduced the severe fruit damages but were not useful to protect against light damages. The experimental shaker produced a higher percentage of slightly damaged oranges. No significant differences in removal efficiency were found between the two harvesting systems. However, removal efficiency using the experimental device could be reduced by 40 percent and working time increase by more than 50 percent when access to the main branches was difficult. In agreement with previous results, the curve representing the branch cumulative removal percentage in time followed a sigmoidal pattern. A model was built showing that during the first 5 s more than 50 percent of the fruits were detached.

**Keywords:** citrus; harvest; vibration; experimental shaker

## 1. Introduction

Citrus production in the Mediterranean region is essential for the agriculture of the coastal countries and decisive for their economies [1]. In Spain citrus production is crucial, Spain is the sixth-largest citrus producer in the world, mostly oranges and tangerines, and most of the production is for fresh consumption. However, the competitiveness of other markets (mainly from the Mediterranean region) is demanding a development of the citrus production system [2].

Improving the cultivation technology has been proposed as a measure to increase the competitiveness of local citrus producers from rural communities [3]. In Spain, production costs, especially labor operations, reduce citrus producer profitability [4]. The Spanish Agriculture Ministry registered a significant increase in citrus production costs in the Valencia region during the period 2011–2017. Several authors propose technological advances to solve the obstacles of the high costs of Spanish fresh citrus production [5–7]. Previous studies have confirmed the inefficiency of farms in conventional systems managing some operations, specifically performing pruning and harvesting operations [8]. Mechanical harvesting systems have been studied to solve the problem of the high costs of manual harvesting [9]. In Spain, mechanical citrus harvesting has also been proposed to reduce high production costs [10]. Mobile platforms to assist harvest operation, where workers place the harvested fruits, have been adapted for citrus crops and studied to improve efficiency

especially when the citrus production is destined for the fresh market [5,11]. Fruit damage could be produced during mechanical and shocking absorbing materials need to be used to reduce fruit impact [12]. Besides, no significant differences in harvest productivity have been proven compared to manual harvest [13]. Two categories of harvesting systems for removing the fruit from the branch were indicated, by human hand or by a mechanical machine or mechanism, and four systems were identified: air shaking, trunk shaking, limb shaking, and canopy shaking [5].

Air systems have been dismissed and have not been studied or developed for harvesting any orchard product. Canopy shakers have been studied for citrus harvesting [14–21]. The force required to remove the fruit was registered (18 percent of the traditionally measured fruit detachment force) and it was suggested that the detachment force was higher inside the canopy than at the edges [22]. Different shaking parameters (frequency and amplitude) were proposed to minimize tree damage and maximize fruit removal using canopy shakers ([23]), rod materials and vibration values (7.5 Hz with an amplitude of 7.6 to 8.9 cm) were suggested. The effect of shaking frequency and penetrating depth on fruit removal were determined [24]. Canopy shakers require an adaptation of the orchard and the trees for mechanical harvesting [25]. Besides, it has been confirmed that fruits receive impacts with other fruit, branches, or machine elements when using canopy shaker systems [26]. Limb and trunk shaking devices have also been studied as vibration systems for mechanical citrus harvesting [9,27–31]. Trunk shakers used to harvest citrus registered frequencies around 15 Hz to 25 Hz and amplitudes around 0.02 to 0.03 m. According to [31], total tree removal percentages registered with the limb shakers (around 50 percent) were lower than with the trunk shakers (between 70 percent and 80 percent).

In the fresh market for citrus from the Mediterranean region, fruit damage should be considered. The possibility of using canvases on the ground and elevated canvases to collect citrus fruits in mechanical harvesting to reduce fruit damages has been previously studied [32]. Besides, mechanical harvesting requires an adaptation to the very low citrus trunk heights. The mechanical citrus harvesting parameters (frequency, amplitude, and duration) to ensure maximum removal efficiency were determined using a laboratory experimental device and comparing the results with field tests [33]. The results showed the possibility of reducing the frequencies and increasing the amplitudes to achieve an optimal mechanical citrus harvest (amplitude values higher than 0.06 m and frequencies around 5 Hz). One of the main problems in citrus mechanical harvesting is the increase of leaf and stem debris [34]. The previous laboratory studies using lower frequencies and higher amplitudes have proven to reduce leaf losses [35].

Conventional trunk shakers can not increase the amplitude to the laboratory proposed values. A lightweight experimental shaker was developed and successfully tested to harvest ornamental citrus trees [36]. The frequency used was between 4 Hz to 6 Hz and the stroke was 0.06 m. The gripper is capable to grasp the low diameter trunk to be shaken. This lightweight experimental shaker was improved to harvest fresh citrus orchards, trees with high trunk diameter, and several main branches. In this case, the main branches need to be grasped due to the high trunk diameter.

The objective of this work was to evaluate the removal efficiency and operational times of the experimental lightweight shaker compared to an orchard tractor-mounted trunk shaker.

## 2. Materials and Methods

### 2.1. Materials

Healthy and well-managed trees (no symptoms of decline, proper leaf and canopy size, and no water stress symptoms) from the variety 'Caracara' (harvesting period from mid-December to mid-January) from an ANECOOP experimental orchard in Museros, Valencia (39º34′10.8» N 0º21′28.8» W) were tested (8–10 February 2021). The trees were planted in a grid with an in-row spacing of 4 m and 5 m between rows. Rows were on trapezoidal ridges (0.3 m in height and 2 m wide at the top). The trunk height was

(mean ± SD) 0.41 ± 0.03 m and the trunk diameter was 0.11 ± 0.01 m. The trees had three to five main branches. The canopy height of the above-ground level was 2 ± 0.2 m. The height from the ground to the canopy skirt was 0.35 ± 0.2 m. The canopy diameter perpendicular to the row was 2.9 ± 0.2 m. The canopy diameter parallel to the row was 3.6 ± 0.2 m.

### 2.2. Harvesting Systems

Two different harvesting systems were tested (experimental lightweight shaker and orchard tractor-mounted trunk shaker). The tractor-mounted trunk shaker was arranged in two parts: one part (with a mass of 640 kg) was attached at the tractor's rear three-point hitch and included the oil tank and the pumps operated by the tractor's power take-off; the other part (with a mass of 730 kg) was coupled to the front three-point hitch and included an extendable arm and a clamp with two moving fingers (Figure 1). Additionally included was the hydraulic motor (vane motor, 80.1 cm$^3$/rev cylinder capacity and 1.27 Nm bar-1 torque) that drives an eccentric mass of 16 kg with an eccentric radius of 0.13 m that produces an orbital vibration (Topavi light shaker, Maquinaria Agrícola Garrido s.l. (Topavi), Autol, Spain, www.topavi.es, accessed on 1 November 2021. In all cases, the tree trunk was clamped at around 0.2 m above the ground. The tractor was a 66 kW four-wheel-drive model.

The experimental lightweight shaker (Figure 2) is a lightweight, linear and low-cost, experimental shaker. The clamp is made of two steel fingers covered with 60 mm thick rubber pads. The fingers were moved by a hydraulic cylinder. The shaker was hitched to the forks of a pedestrian hydraulic tractor (Hinowa, Nogara, VR, Italy; www.Hinowa.com, accessed on 1 November 2021, 'Hinowa' model 'HS 11000' provided with a fork elevator. The shaker was powered by a hydraulic motor, that received the oil from the external supplies of a Lamborghini Plus 990F tractor that gave a flow of 21 L per min at 100 bar, or from a John Deere 5820 tractor that gave a flow of 26 L per min at 100 bar. The vibration amplitude used was 0.60 m.

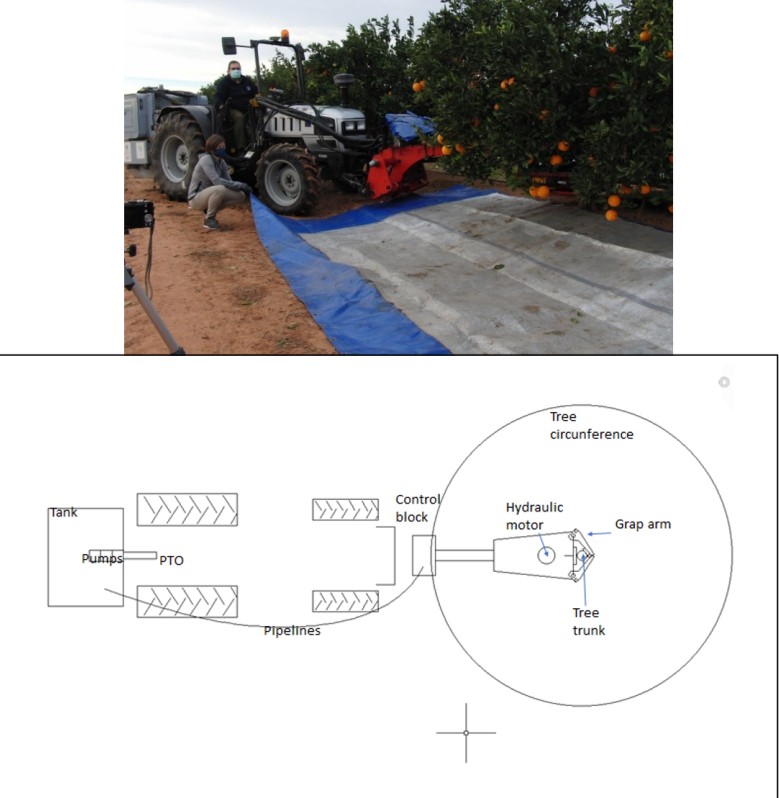

**Figure 1.** Tractor and trunk shaker used in the study (**top**) and a schematic diagram (**bottom**).

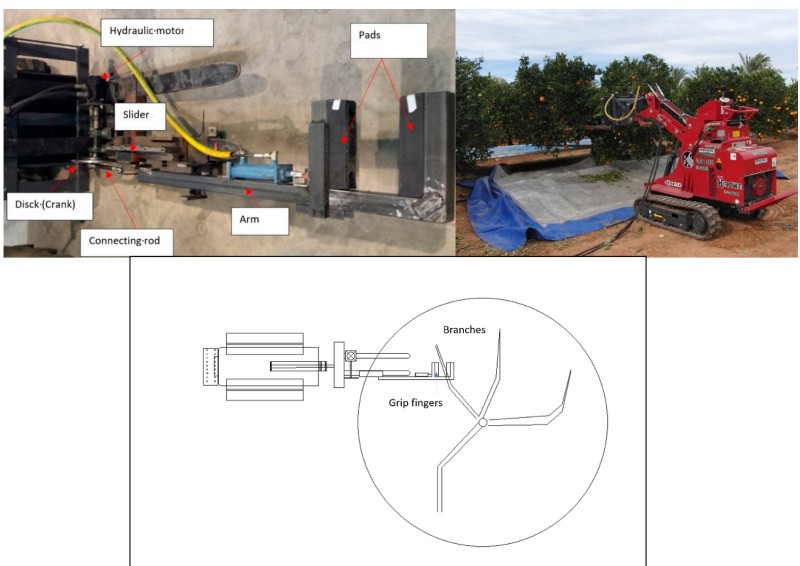

**Figure 2.** Experimental light-weight shaker elements (**left**, **top**), the shaker vibrating a tree (**right, top**) and a schematic diagram (**bottom**).

### 2.3. Methodology

In order to reduce the tree effect, nine trees (with similar tree structures) were selected and tested (three with the trunk shaker and six with the experimental shaker). With the experimental shaker between two to four main branches were shaken per tree. With the trunk shaker, two strokes of less than 4 s each of vibration time were applied. Fifty fruits from each harvesting system (experimental shaker and trunk shaker) and collecting system (ground, ground covered with canvases, elevated canvases, and manually collected (Figure 3) were carefully taken to the laboratory after harvesting and were stored at room temperature with high relative humidity (24 degrees and 90 percent) for fourteen days in order to increase bruise visualization. In the case of the elevated canvases, the number of samples was duplicated. Besides, sixty fruits were manually collected, thirty to analyze fruit damages and thirty to measure fruit weight and maturity index. Fruit damage was evaluated by measuring the percentage of slightly damaged fruits (small cuts, rubbing, and very small bruises) and the percentage of considerably damaged fruits (bruises and cut wounds). Slightly and severely damaged fruit percentages were evaluated according to the local market standards and based on [32].

The detachment percentage was calculated by counting and weighing the oranges removed and those remaining on the trees after being picked by hand. To determine the removal percentage according to the vibration time, high-speed video recordings were analyzed. The accumulated number of detached fruits according to the total number of fruits was calculated and related to the vibration time. The actual frequencies were measured using a triaxial accelerometer and recorder (Gulf Coast Data Concepts, LLC, Waveland, MS, USA; http://www.gcdataconcepts.com, accessed on 1 November 2021. that recorded the vibrations at 400 Hz and with high-speed 300 fps video recording (counting cycle time). The video recording was also used to calculate the different working and access times.

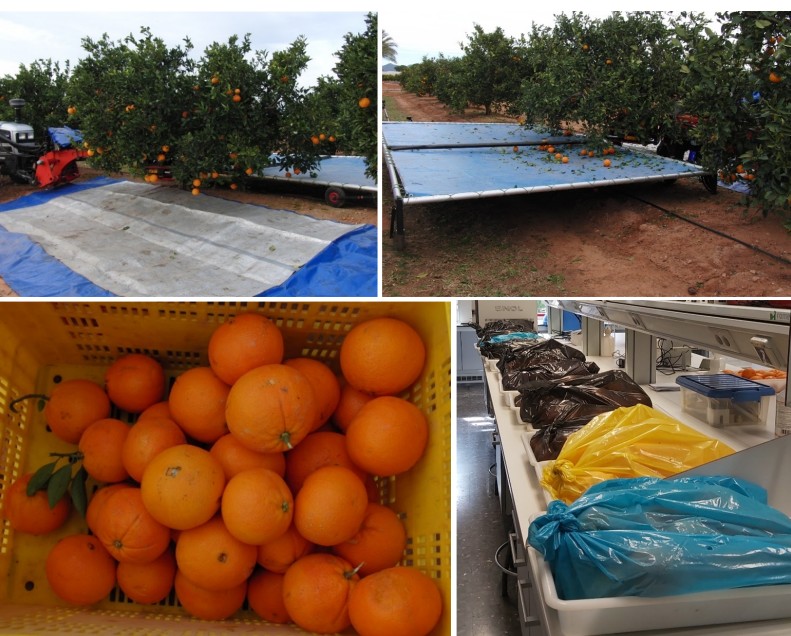

**Figure 3.** Collecting canvases (ground covered with canvases (**left**, **top**), elevated canvases (**right**, **top**), collected fruit (**left**, **bottom**) and fruit stored in the laboratory at room temperature and high humidity (**right**, **bottom**).

## 3. Results

### 3.1. Fruit Damage

As was expected, a higher percentage of damaged fruits was obtained when the fruits were collected without a protection canvas (Figure 4). The elevated canvas reception produced lower severe damage than the fruits manually harvested and a little higher percentage of slightly damaged fruits. However, the canvases on the ground protected against severe damage but produced a higher percentage of slightly damaged fruits (similar to the fruit collected directly on the ground).

In order to analyze the citrus damage produced during the vibrating time, before collecting, only the fruits collected with the elevated canvases were studied. When comparing the considerably damaged fruits using both harvesting systems no significant differences were found. However, the percentage of slightly damaged fruits was substantially higher when using the experimental shaker with high amplitude compared to the traditional trunk shaker ($p$-value = 0.0735 in the ANOVA analysis studying the effect of the harvesting system on the slightly damaged fruit percentage) (Figure 5). In this line, 74 percent of the slight damages in fruits detached with the experimental shaker were small cuts and rubbings compared to the 14 percent in the fruits detached with the trunk shaker. When analyzing the videos recorded during the shaking time, a high amplitude movement of the fruits related to the high amplitude of the branch vibration is confirmed. Some of the fruits, during the vibration, are rubbed against the branches and other fruits.

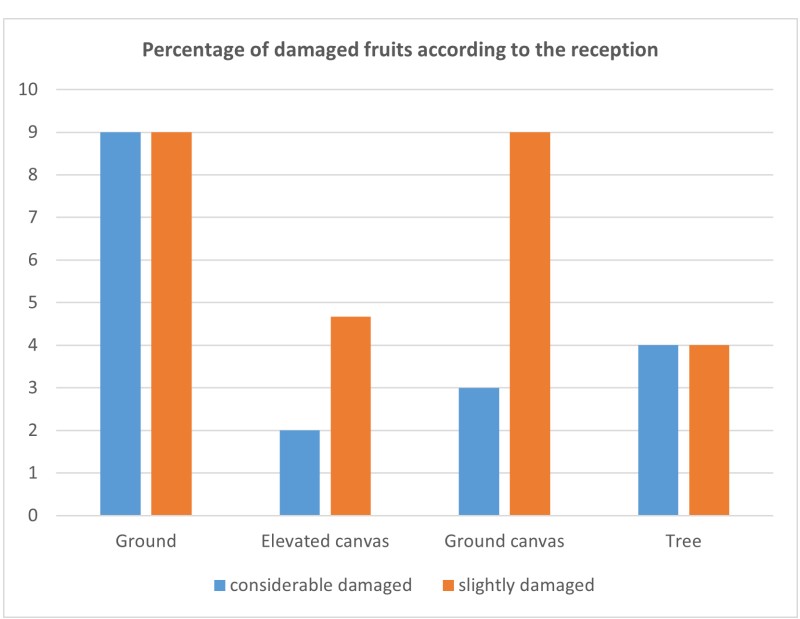

**Figure 4.** Percentage of damaged fruits (considerable damaged and slightly damaged) according to the reception system.

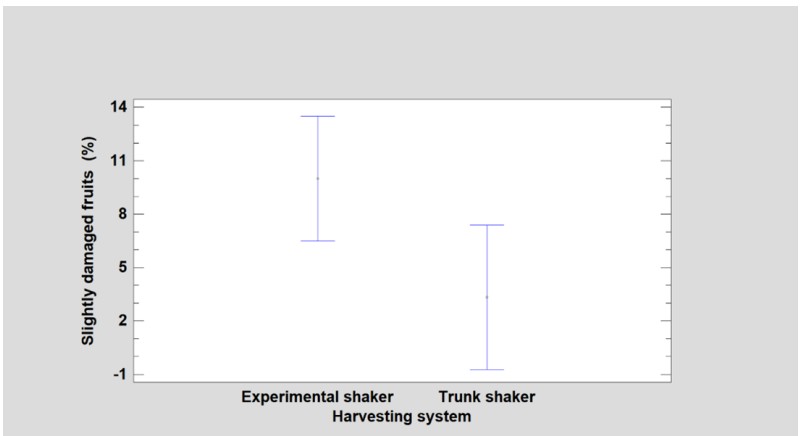

**Figure 5.** Slightly damaged fruits (percentage) related to the harvesting system (average value and Fisher LSD interval), only the ones collected on the elevated canvases.

### 3.2. Detaching Times

The registered frequency of the trunk shaker was around 19 Hz with an accumulated time lower than 8 s, with two shakes of less than 4 s each of vibration time. The experimental shaker worked with a lower frequency (5.7 Hz) with an accumulated time of 50 s. The working time increased when the access to the main branches to be shaken was difficult. The time required to access the vibrated branches were between 16 s to 52 s (37.8 s ± 19.2 s). However, some of the main branches had extremely difficult access and the required time was not included. The gripping time was between 3 s to 5 s, and the shaking time was between 14 s to 50 s (31.3 s ± 10.5 s).

### 3.3. Removal Percentage

No significant differences were found between the removal percentage using the experimental shaker and the trunk shaker (69.3 percent ± 8.1 percent compared to 77 percent ±1.6 percent, respectively), Figure 6, ($p$-value = 0.4076 in the ANOVA analysis). However, the removal percentage of the experimental shaker is lower if the total removal percentage of the tree is considered (including the fruit from the non-vibrated branches).

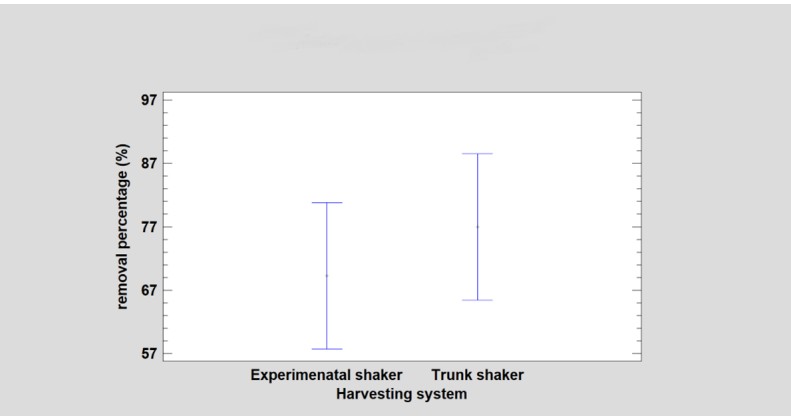

**Figure 6.** Removal percentage (percentage) using the experimental shaker and the trunk shaker (average and Fisher LSD intervals).

### 3.4. Branch Removal Percentage According to Time

The branch removal percentage-related time was analyzed using the vibration recorded videos. As reported in previous laboratory tests using low frequency and high amplitude [33], the removal percentage change follows a sigmoidal behavior with the vibration time (average $R^2 = 81$). Figure 7 shows, as an example, the accumulated removal percentage according to time in six different branches with the $R^2$ value of the fitting curve.

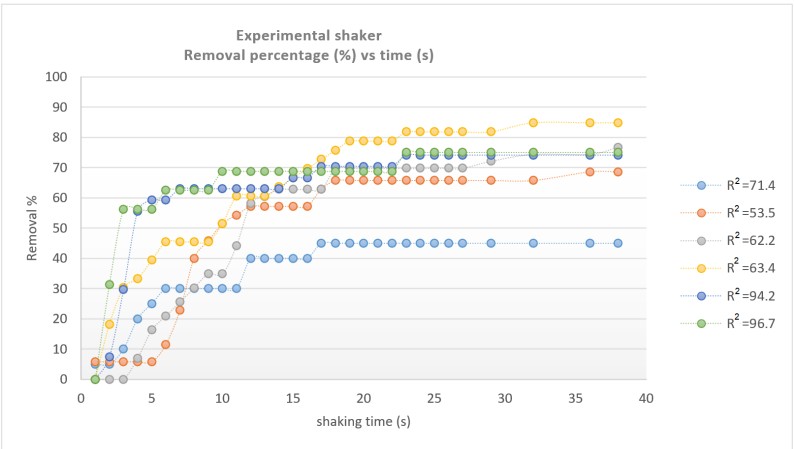

**Figure 7.** Accumulated branch removal percentage (percentage) according to time using the experimental shaker in six different branches ($R^2$ value of the fitting sigmoidal curve).

Using the data of the accumulated removal percentage related to time from the different vibrated branches, the average sigmoidal curve parameters were calculated and a modelized curve was built, Figure 8. In the first 5 s, 50 percent of the fruit from the branch was removed, after 12 s 60 percent of the fruit and after this time only 4 percent more was removed.

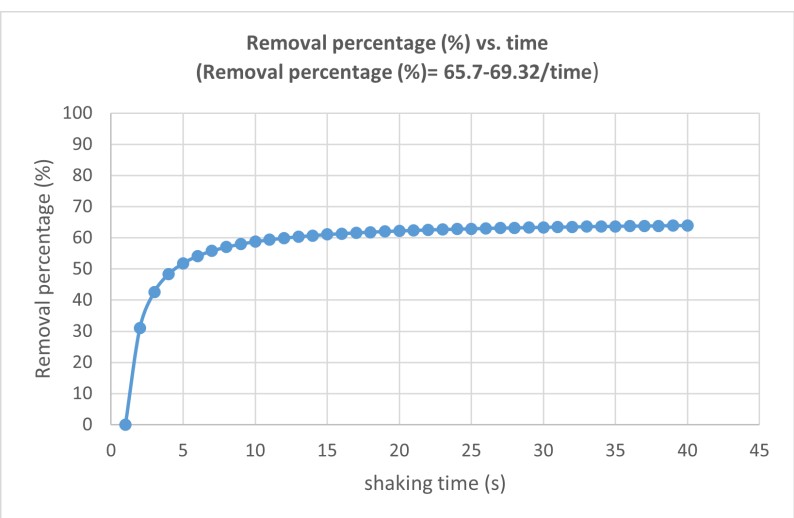

**Figure 8.** Modelized curve of the branch removal percentage related to time.

## 4. Discussion

Fruit mechanically harvested with the experimental shaker presented adequate qualities for fresh market. The elevated canvases collected the fruit without damage and only a low percentage of fruits were detached without calyx. The percentage of slightly damaged fruits is higher when using the experimental shaker compared to the trunk shaker. This fact could be due to the higher amplitude of the vibration, producing a higher amplitude in the fruit movement and favoring frictions of the fruits against branches and other fruits. This fact needs to be addressed in further studies.

The experimental shaker works vibrating the main branches of the tree. When considering only the vibrated branches, no significant differences were found in removal percentage between the experimental shaker and the trunk shaker (69 percent compared to 77 percent, respectively). However, some of the branches are not accessible to be shaken, reducing the real removal percentage of the experimental shaker. This total tree removal percentage depends on the structure of the tree. Removal efficiency using the experimental device could be reduced by 40 percent and working time increased by more than 50 percent when access to the main branches was difficult. A previous pruning system focused on three accessible main branches would improve the total tree removal percentages. An adequate pruning system would also reduce the time required to access the branches and reduce working times.

It has been proven that removal percentage related to time follows a sigmoidal behavior with the experimental shaker as it was previously proven with trunk shakers. In the first seconds, most of the fruit is already removed and continuing with the shaking does not increase the removal percentage and could produce negative effects. With the trunk shaker, almost all the detached fruit fell in the first 2 to 3 s of shaking. However, with the experimental shaker, the time required to detach the fruit is longer, around 12 s.

## 5. Conclusions

The canvases on the ground used to collect the oranges reduced the severe damages but were not useful to protect against light damages. However, the capability of the elevated canvases to reduce damages was verified. It seems that the high amplitude vibration of the experimental shaker produced a higher percentage of slightly damaged oranges, especially little cuts and abrasions. The higher damage percentage could be due to the fact that some of the fruits are being rubbed against the branches and other fruits during the vibration. However, further research should be carried out to study other factors involved in fruit injuries previous to the collecting systems during mechanical harvesting. The trunk shaker frequency was around 19 Hz and with an accumulated time lower than 8 s, with two shakes of less than 4 s each of vibration time. The experimental shaker worked with a lower

frequency (5.7 Hz) and with an accumulated time of 50 s. However, it was proven that the first 12 s were the most important. A high amplitude–low frequency experimental citrus branch shaker was tested in field conditions. The removal percentage (considering the vibrated branches) is slightly lower using the experimental shaker compared to the trunk shaker (69 percent compared to 77 percent). With the experimental shaker, the differences in removal percentage are related to the impossibility of vibrating some of the main branches This fact and the difficulty accessing some of the main branches reduced the working capacity of the experimental shaker. The removal percentage with the experimental shaker follows a sigmoidal behavior with the vibration time. Most of the fruit was removed in the first 12 s.

**Author Contributions:** Conceptualization, A.T. and C.O.; methodology, C.O.; software, C.O. and A.T.; validation, C.O.; formal analysis, C.O., A.T. and S.C.-G.; investigation, C.O., A.T. and S.C.-G.; resources, C.O., A.T. and S.C.-G.; data curation, C.O.; writing—original draft preparation, C.O.; writing—review and editing, C.O., A.T. and S.C.-G.; visualization, C.O., A.T. and S.C.-G.; supervision, C.O., A.T. and S.C.-G.; project administration, C.O.; funding acquisition, C.O., A.T. and S.C.-G. All authors have read and agreed to the published version of the manuscript.

**Funding:** This research has been fund by the European Agricultural Fund for Rural Development and cofounded by the Ministerio de Agricultura, Pesca y Alimentación (project GO "Avances tecnológicos para la modernización y la sostenibilidad en la producción de cítricos CITRUSTECH").

**Institutional Review Board Statement:** Not applicable.

**Informed Consent Statement:** Not applicable.

**Conflicts of Interest:** The authors declare no conflict of interest

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
