# Peer review of "Comparison of a Lightweight Experimental Shaker and an Orchard Tractor Mounted Trunk Shaker for Fresh Market Citrus Harvesting"

_agriculture, doi:10.3390/agriculture11111092_

Round 1

Reviewer 1 Report

In this paper, the principles of the two harvesting systems need to be introduced in more detail. The figure 1 and Figure 2 are not enough, and some schematic diagrams of the two harvesting systems should be provided.

The experimental results need to be explained and discussed in more detail.

Author Response

Thank you very much to the reviewer for the helpful comments and suggestions.

The reviewer's comments have been considered carefully, and the manuscript has been extensively revised. The modification in text have been highlighted in yellow. And the responses to the reviewer have been carefully considered

Reviewer 1

In this paper, the principles of the two harvesting systems need to be introduced in more detail. The figure 1 and Figure 2 are not enough, and some schematic diagrams of the two harvesting systems should be provided.

- The literature review about the principles of the harvesting systems has been improved (more than thirty new references have been added), new lines 17-72.

- The two harvesting systems have been described with more detail. Figure 1 and Figure 2 have been improved adding the schematic diagrams.

The experimental results need to be explained and discussed in more detail.

- The experimental results have been explained in a discussion section that has been added (new lines 190-214).

Reviewer 2 Report

In this manuscript, the performance of two different shaker was compared for citrus harvesting; light-weight experimental shaker and an orchard tractor mounted trunk shaker. The main idea of study is interesting and authors approach is scientifically good. There are some minor issues that should be considered. Having stronger literature review and focusing more on previous relevant researches are highly recommended, in addition to add nomenclature table.  

Line 9: “No significant differences in removal efficiency were found between the two harvesting systems”. In which level, there was no significant differences?

Line 10: “removal efficiency using the experimental device was reduced and working time increased when the access to the main branches was difficult.”. The percentage of reducing removal efficiency and increasing time should be added.  

Line 39: The location (latitude, Longitude), date/time of experiment and normal date/time of harvesting in the location should be mentioned. In addition, the criteria for considering to evaluate the tree health and condition should be added.

Line 54: Information about hydraulic motor is needed.

Line 70: It is good to add the picture of all collected samples

Line 99: statistical analysis and level of evaluation should be added.

Author Response

Thank you very much to the reviewer for the helpful comments and suggestions.

The reviewer's comments have been considered carefully, and the manuscript has been extensively revised. The modification in text have been highlighted in yellow. And the responses to the reviewer have been carefully considered

In this manuscript, the performance of two different shaker was compared for citrus harvesting; light-weight experimental shaker and an orchard tractor mounted trunk shaker. The main idea of study is interesting and authors approach is scientifically good. There are some minor issues that should be considered. Having stronger literature review and focusing more on previous relevant research are highly recommended, in addition to add nomenclature table.  

- The literature review has been substantially improved (more than thirty new references related to citrus harvesting background and mechanical harvesting systems have been added based on previous relevant research), new lines 17-72.

Line 9: “No significant differences in removal efficiency were found between the two harvesting systems”. In which level, there was no significant differences?

- The level in which there were no significant differences has been included (P value=0.4076 in the ANOVA analysis), new line 174.

Line 10: “removal efficiency using the experimental device was reduced and working time increased when the access to the main branches was difficult.”. The percentage of reducing removal efficiency and increasing time should be added.

-  The percentage of reducing removal efficiency and increasing time have been included, new line 11 and in the lines 202-204 in the new discussion section.

Line 39: The location (latitude, Longitude), date/time of experiment and normal date/time of harvesting in the location should be mentioned. In addition, the criteria for considering to evaluate the tree health and condition should be added.

- The information about location (latitude, Longitude), date/time of experiment and normal date/time of harvesting in the location have been added, new lines 84-86.

- No actual measurements were carried out to evaluate the tree health. However, a citrus field technician evaluated the condition of the selected trees based on symptoms of decline, proper leaf and canopy size and  water stress symptoms, new lines 83-84.

Line 54: Information about hydraulic motor is needed.

- The information about the hydraulic motor has been included, news lines 101-102.

Line 70: It is good to add the picture of all collected samples.

- A picture of the collected samples has been added in the Figure 3.

Line 99: statistical analysis and level of evaluation should be added.

- The statistical analysis parameters and the level of evaluation has been included, new lines 173-174 (about removal percentages), new lines 181-183 and Figure 7(about Branch removal percentage according to time).

Reviewer 3 Report

It is an interesting paper. The use of shaker harvesting fresh market citrus is a good idea.

  1. They should provide assessment methods for severe damage and slightly damage.
  2. Only nine trees were tested and only six branches were shook by the experimental shaker. I was perplexed in the manuscript by the lack of test samples.
  3. I was also perplexed in the manuscript by the lack of appropriate statistical summary statistics such as Standard deviation, R2 and RMSEP.
  4. There are many interesting researches in citrus harvesting and trunk shaker harvesting in recent years. In introduction chapter, they could include more recently published references in citrus harvesting and trunk shaker harvesting areas.
  5. Did they consider the effect of fruit tree parameters (such as height, trunk diameter) on the experimental results?
  6. Results are interesting but their discussion is necessary.

Author Response

Thank you very much to the reviewer for the helpful comments and suggestions.

The reviewer's comments have been considered carefully, and the manuscript has been extensively revised. The modification in text have been highlighted in yellow. And the responses to the reviewer have been carefully considered

It is an interesting paper. The use of shaker harvesting fresh market citrus is a good idea.

1. They should provide assessment methods for severe damage and slightly damage.

- The assessment method has been provided, new lines 130-131.

2. Only nine trees were tested and only six branches were shook by the experimental shaker. I was perplexed in the manuscript by the lack of test samples.

- In order to reduce the tree effect, from a sample of more than thirty trees, nine trees (with similar tree structure) were selected and tested (three with the trunk shaker and six with the experimental shaker), based on the criteria of a citrus field technician. With the experimental shaker between two to four main branches were shaken per tree.

3. I was also perplexed in the manuscript by the lack of appropriate statistical summary statistics such as Standard deviation, R2 and RMSEP.

- The statistical analysis parameters and the level of evaluation has been included, new lines 173-174 (about removal percentages), new lines 181-183 and Figure 7(about Branch removal percentage according to time).

4. There are many interesting research in citrus harvesting and trunk shaker harvesting in recent years. In introduction chapter, they could include more recently published references in citrus harvesting and trunk shaker harvesting areas.

- The literature review has been substantially improved (more than thirty new references related to citrus harvesting background and mechanical harvesting systems have been added based on previous relevant research), new lines 17-72.

5. Did they consider the effect of fruit tree parameters (such as height, trunk diameter) on the experimental results?

- The tree effect was reduced selecting trees with similar structure. However, trunk diameter, trunk height and canopy dimensions were measured, no relation with the harvesting parameters was found.

6. Results are interesting but their discussion is necessary.

- A discussion section has been added (new lines 190-214).

Round 2

Reviewer 1 Report

The issues raised  have be addressed very well.

Reviewer 3 Report

No.